# LEGO® as a versatile platform for building reconfigurable low-cost lab equipment

**Diane N. Jung**[1], **Kailey E. Shara**[2], **Carson J. Bruns**[1,2]*

1 Paul M. Rady Department of Mechanical Engineering, University of Colorado Boulder, Boulder, Colorado, United States of America, 2 ATLAS Institute, University of Colorado Boulder, Boulder, Colorado, United States of America

* Carson.Bruns@colorado.edu

## Abstract

Laboratory equipment is critical for automating tasks in modern scientific research, but often limited by high costs, large footprints, and sustainability concerns. Emerging strategies to provide low-cost research automation tools include microfluidic devices, open-hardware devices, 3D printing, and LEGO® products. LEGO®-based equipment may be advantageous with respect to sustainability, since their inherent modularity enables disassembly, re-purposing and re-use. To explore the feasibility and cost savings of replacing conventional lab equipment with LEGO®-based alternatives, we developed and characterized the performance of three LEGO® Technic™ laboratory tools: a syringe pump, an orbital shaker, and a microcentrifuge. These three machines share 384 pieces in common and can be constructed in series (687 pieces, <$83 USD) or in parallel (1215 pieces, <$174 USD). As a proof of concept, calcium carbonate microparticles were synthesized and isolated using the LEGO®-based and analogous commercial equipment, yielding comparatively similar results. Moreover, the ability to program custom shake profiles for the LEGO®-based orbital shaker gave access to a wider range of particle characteristics than the commercial shaker. We propose that the high cost savings and reusability of LEGO®-based lab tools extends beyond their well-established efficacy in K-12 STEM education to an attractive resource for budget-, space- and/or sustainability-conscious laboratories.

## Introduction

Machines that automate procedural operations are ubiquitous and essential in most experimental labs. In biochemistry, for example, instrumentation is essential for countless experiments involving spectrophotometry, centrifugation, electrophoresis, microscopy, etc. [1]. In chemical synthesis, equipment such as heat exchangers, stir plates, vacuum pumps, and rotary evaporators have been indispensable reaction processing tools for many decades [2], while relatively newer tools such as automated and high-throughput chromatography systems are becoming widespread [3]. Still, our own observations [4] of synthetic chemists working in the laboratory identified numerous tasks associated with substantial physical and cognitive burden, some of which could potentially be mitigated by appropriate automation tools. New analytical and processing tools, from robotic liquid handlers [5] to solvent extraction systems [6],

**Data availability statement:** All relevant data are within the paper and its Supporting Information files.

**Funding:** U.S. National Science Foundation (NSF) Award No. 2222952. Collaborative Research: FW-HTF-R: RoboChemistry: Human-Robot Collaboration for the Future of Organic Synthesis. The funders had no role in study design, data collection and analysis, decision to publish, or preparation of the manuscript.

**Competing interests:** The authors have declared that no competing interests exist.

continue to improve experimental research efficiency [7]. Though specific equipment needs vary, almost all labs make use of machines, yet most laboratory equipment is expensive and specialized for specific subtasks. Robotic solutions have been developed to overcome the narrow scope of laboratory instrumentation and provide more generalized support through end-to-end automation of an entire chemical reaction [6,8,9], as well as machine learning algorithms for real-time synthesis monitoring and self-optimization [8,10,11]. However, budget- and space-constrained laboratories may be unable to purchase high-cost automation systems and would benefit instead from small, low-cost, reconfigurable laboratory equipment. Reconfigurable equipment may also help mitigate the stockpiling of obsolete equipment in storage and landfills [12].

Application of emerging technologies to lab equipment can provide more diverse and accessible automation tools. For example, microreaction technology [13] such as microfluidic "lab-on-a-chip" (LOC) devices lower the cost of reagents and instrumentation by reducing the scale of experiments [14,15], enhance efficiency through automated liquid handling in biological assays [16,17], and provide inexpensive point-of-care diagnostics [18,19]. Owing to the high fabrication costs of LOC devices, recent research has turned to more economical fabrication techniques [20]. Open-hardware solutions can also help reduce research costs [21,22] by integrating various manufacturing methods such as 3D printing, laser cutting, and CNC milling with off-the-shelf electronic components and open-source microcontrollers [23–26], but assembly and operation may require specialized knowledge or extensive support from online communities [27–29]. 3D printing has become a mainstream solution for low-cost, customizable laboratory equipment, as researchers can download open-source designs from online repositories and modify them for their needs or design their own custom parts with online support [29]. Diverse 3D-printed tools have been validated, from chemistry reactionware [30] and custom lab equipment [31] to medical models, hygiene materials, and rehabilitation equipment [32]. Although 3D printing is affordable and customizable, limitations include (i) upfront investment in the necessary tools, (ii) time required for training and practice, and (iii) a need for separate assembly and integration of non-printable parts such as electronic components and microcontrollers [33].

LEGO® parts have been used broadly in scientific research because they (i) can be assembled and disassembled easily outside of a machine shop without specialized equipment or training, (ii) can be reconfigured and reused as needed, (iii) do not require significant upfront investment, (iv) have components and controls that can be integrated into a unified user-friendly platform, and (v) are amenable to sterilization [34,35]. Examples of LEGO®-based applications in scientific research include building prototypes to validate designs [34,36], autonomous mobile [37] or cable-driven parallel [38] robots, and custom parts for experimental procedures such as a sensor holders [39], cm-scale liquid reservoirs [40], mechanical cell stimulators [35], and magnetic resonance imaging (MRI) phantoms [41]. LEGO® robots have also been used in research involving trajectory tracking [42], development of route planning models [43], and design of a control program for robots [44]. Some researchers have also reported low-cost LEGO®-based laboratory equipment such as an absorption spectrophotometer [45], 3D bioprinter [46], gradient mixer [47], syringe pump [48], and an automated pipetting robot [49]. In this paper, we developed several LEGO®-based laboratory automation tools – a syringe pump, orbital shaker/reaction agitator, and microcentrifuge, each with an on-board graphical user interface (GUI) in the form of a LEGO® EV3 Intelligent Brick programmed with LEGO® MINDSTORMS® EV3 software – to investigate their feasibility and cost savings as alternatives to conventional lab equipment. We show that these devices can be assembled and reconfigured at low cost and offer satisfactory performance in a demonstration of calcium carbonate ($CaCO_3$) microparticle synthesis and isolation, supporting

the feasibility of LEGO®-based automation tools for resource-constrained or sustainability-conscious labs.

## Materials and methods

**Materials and instrumentation.** 2-(Dimethylamino)ethyl acrylate (DMAEA), poly(ethylene glycol) diacrylate ($M_n$ 700, PEGDA), 2-hydroxy-2-methylpropiophenone (Darocur), $Na_2CO_3$, and heavy mineral oil (Ward's Science) were purchased from Fisher Chemical or Sigma-Aldrich and used without further purification. $CaCl_2$ was purchased from Pure Original Ingredients. All LEGO® pieces were purchased on bricklink.com. The LEGO® MINDSTORMS® EV3 Home Edition software was downloaded for free on the LEGO® website and was used to program the robot and develop the GUI. Assembly instructions were developed using BrickLink Studio, which was downloaded free of charge on bricklink.com. The volume-weighted distributions of hydrodynamic particle diameters were estimated by dynamic light scattering using a Malvern Panalytical Mastersizer 3000, where the dried $CaCO_3$ samples were suspended into the dispersion unit filled with milli-Q® water, to a laser obscuration level of ~4%. SEM images were taken at 15 kV with a 30 mm spot size on a Hitachi SU3500 VP Scanning Electron Microscope; samples were prepared from a suspension of $CaCO_3$ microparticles in ethanol (~5 mg/mL) that had been sonicated at 40 kHz (Fisher Scientific FS30 Ultrasonic Cleaner) for 5 min, then drop-cast on carbon tape, dried overnight, and coated with 4 nm of platinum (Cressington 108 Auto/SE Sputter Coater) under vacuum.

**Syringe pump testing.** To characterize the dispense rates of the syringe pump at different EV3 motor settings, three disposable syringes of different volumes: 5 mL (Fisherbrand Luer-Lock Syringe), 10 mL (BD Luer-Lock Syringe), and 12 mL (A AKRAF 12 mL Plastic Syringe) were filled with water, loaded onto the syringe pump, and dispensed at motor speed settings of 1, 5, or 9. The water was dispensed into a weigh boat placed on an analytical balance (RAD-WAG AS 60/220.R2) and the weight was recorded every 30 seconds for 5 consecutive minutes. This procedure was repeated three times and the recordings at each time interval were averaged. The average mass of dispensed water was converted to volume, assuming a density of 1 g/mL, to determine dispense rates from the slope of the linear fit in a graph of volume dispensed vs. time elapsed.

**Orbital shaker testing.** The time-resolved revolution data for the orbital shaker motor was captured with the Data Streamer function in Microsoft Excel on a PC via a SparkFun Redboard, an Arduino-compatible development board. The LEGO® EV3 cable that connects the EV3 brick to the EV3 large motor was cut and the wires were exposed and then re-attached with just the ground (red), blue, and yellow wires (quadrature encoder data lines) connected to the Redboard (red to ground, blue to pin 3, yellow to pin 2). The Redboard was used to capture the quadrature encoder data and send it to the PC via USB. Rotational speed was calculated through collection of five video recordings and visualized as box plots.

**Microcentrifuge testing.** The rotational speed and consistency of the microcentrifuge was measured using a photogate system, assembled using an infrared (IR) emitter, an IR detector, and an Arduino Uno. A code was written to read data from the IR detector in real time and the signal was captured with a serial port terminal application, CoolTerm, on a PC via Arduino Uno board. Whenever a microcentrifuge arm passed through the detection area between the emitter and detector, a disturbance in the signal was recorded. We obtained signal data for ~10 s every minute during 5 consecutive minutes of centrifugation. Validation of the LEGO® microcentrifuge was carried out by centrifugation of microgels. Polyacrylamide-based microgels were fabricated by photo-crosslinking a pre-gel solution containing DMAEA (20% w/w), PEGDA (40% w/w), Darocur (10% w/w), and degassed water (30% w/w) under

a contact photolithography setup [50]. The size and shape of the microgels were defined by a photomask with square transparencies (200 $\mu$m feature size) and scotch tape spacers of 100 $\mu$m thickness. The pre-gel solution was cured under ultraviolet (UV) light (Digikey, $\lambda$ = 365 nm) at 10 mW/cm$^2$ for 10 s. 1 mL of this solution was poured into 1.5 mL microcentrifuge tubes. One of the microcentrifuge tubes were centrifuged at ~630 rpm / 30 rcf in the LEGO® microcentrifuge for 5 min, while a control tube was left standing for 24 h of gravitational settling time.

**CaCO$_3$ microparticle synthesis.** CaCO$_3$ microparticles were synthesized based on a published procedure[51] that was modified to utilize syringe pumps and orbital shakers. 2 mL of 0.33 M Na$_2$CO$_3$ solution in a 10 mL disposable syringe was dispensed at a 2.42 mL/min dispense rate using a syringe pump into a 20 mL scintillation vial containing 2 mL of 0.33 M CaCl$_2$ that was being agitated at ~60 rpm on an orbital shaker. After dispensing was complete, the scintillation vial was capped and agitated at ~210 rpm for 10 min. This synthesis procedure was duplicated using both a conventional syringe pump (SONO-TEK 12-05126 Dual Syringe Pump) and orbital shaker (KJ-201BD), as well as the LEGO®-based syringe pump and orbital shaker. A third synthesis was performed in the same manner, except the final 10 min agitation employed the custom-programmed mixed shake profile of the LEGO® orbital shaker at the maximum speed setting. For centrifugation, 1 mL aliquots of the stirring crude reaction mixture were transferred into 12 1.5-mL microcentrifuge tubes and centrifuged for 5 min at maximum speed in either a commercial Fisherbrand (6000 rpm / 2000 rcf) or LEGO® (630 rpm / 30 rcf) microcentrifuge. The supernatant was removed using a micropipette and all samples were placed in a vacuum oven at 40°C for ~24 h to remove residual water. Yield was calculated from the mass of the dried product using an analytical balance (RADWAG AS 60/220.R2). Dry CaCO$_3$ samples required for particle size and morphology analysis were obtained by air-drying the crude reaction mixture in a petri dish overnight.

**Statistical analysis.** The mean and standard deviation were calculated for (i) 3 measurements to characterize syringe pump volume dispensed vs. elapsed time (Fig 1D, E, and F), (ii) 6 samples to calculate CaCO$_3$ microparticle yield, and (iii) 5 readings of CaCO$_3$ microparticle volume-weighted hydrodynamic diameter (Fig 4E). Box plots of the orbital shaker / reaction agitator rotational speed vs. time and rotational speed vs. weight were based on five measurements (Fig 2F and G).

## Results

We designed and constructed a LEGO®-based syringe pump, orbital shaker, and microcentrifuge, then validated their performance in comparison with conventional lab equipment in a CaCO$_3$ microparticle synthesis, and performed a comparative cost analysis on these equipment.

### A LEGO® syringe pump

Pumps are devices frequently used to dispense precise and accurate fluid volumes at a controlled rate in laboratory settings such as chemical reactions that require gradual addition of reagents [52], analytical techniques including mass spectrometry and liquid chromatography, and precisely controlled drug infusion [53]. While many methods of fluid pumping are available to meet a wide range of needs, not all pump types are conducive to LEGO®-based construction; for example, the flexible membranes of diaphragm pumps and the microstructured components of micropumps [54] are not available in the LEGO® catalog. A syringe pump, however, is a simple and versatile tool that operates by mechanically pushing a piston with

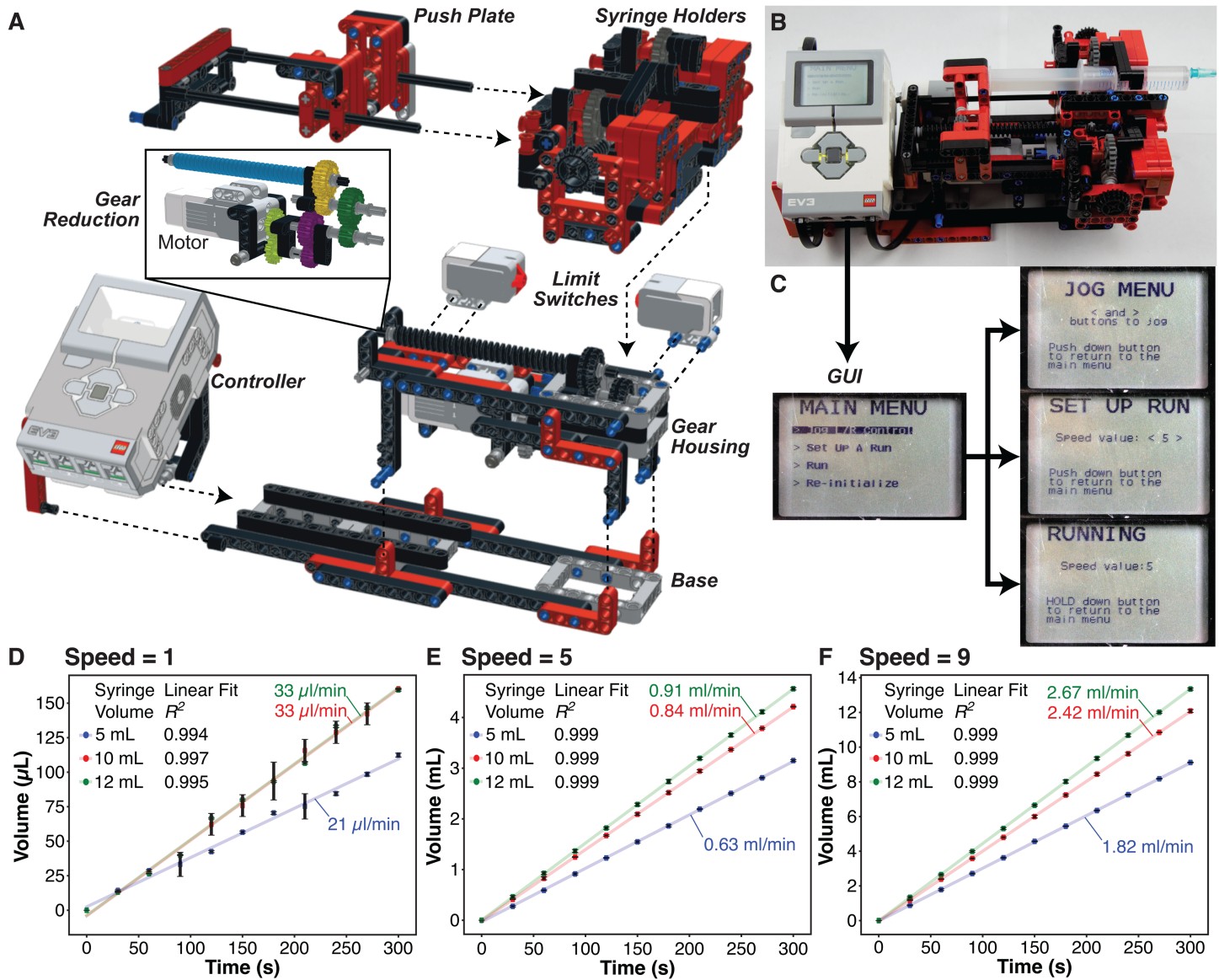

**Fig 1. A LEGO® syringe pump.** (**A**) Exploded oblique view of the sub-assemblies of a LEGO®-based syringe pump. (**B**) Photograph of the fully assembled LEGO® syringe pump. (**C**) Graphical User Interface (GUI) menu flowchart. (**D**) Plot of volume dispensed vs. elapsed pumping time for syringe volumes of 12 mL (green), 10 mL (red), and 5 mL (blue) at speed setting 1. (**E**) Plot of volume dispensed vs. elapsed pumping time for syringe volumes of 12 mL (green), 10 mL (red), and 5 mL (blue) at speed setting 5. (**F**) Plot of volume dispensed vs. elapsed pumping time for syringe volumes of 12 mL (green), 10 mL (red), and 5 mL (blue) at speed setting 9. The standard deviation of the mean at each data point, averaged over 3 trials, is represented by black error bars.

a stepper motor to withdraw or eject fluid from a syringe. Syringe pump prices range from ~150 USD (used) to ~6,200 USD (new), based on prices observed in various online marketplace listings in November 2024. Most syringe pumps can accommodate one or two syringes, while some specialized syringe pumps are available for up to twelve syringes.

We designed a syringe pump (Fig 1) that can be assembled (instructions are provided in the Supplemental Information) out of a total of 505 LEGO® Technic™ pieces, costing ~67 USD (based on lowest prices on bricklink.com, June 2024). The assembled syringe pump (Fig 1B) has dimensions of 25 × 20 × 13 cm and possesses five sub-assemblies (Fig 1A):

syringe holders, push plate, motor and gear reduction, gear housing (with limit switches), and a base. The syringe pump holder can accommodate up to two syringes of outer diameters ranging from ∼13.5 mm to ∼16.5 mm, corresponding to disposable syringe volumes ranging from 5 mL to 12 mL. The pusher block is driven by a gear system attached to a LEGO® EV3 Medium Servo Motor, which is specified to reach up to 250 rpm with a running torque of 8 N/cm and a stall torque of 12 N/cm. The EV3 Intelligent Brick that powers and controls the motor is a Linux-based computer with a six-button interface and an illuminated 178 × 128 pixel display, where the on-board GUI (Fig 1C) is accessed. The GUI allows the user to (i)"jog" or manually drive the pusher block forward and backward, (ii) set up a dispensing rate on a numerical scale of 1–9 (which can be translated to volume-based rates with calibration), (iii) see the status of the pump while it is running, or (iv) re-initialize, which automatically drives the pusher block to the its minimum and maximum locations, constrained by limit switches in the hardware, to determine the range of the pusher block and when the run is complete. The code for the GUI was written in LEGO® MINDSTORMS® EV3 Home Edition software.

We calibrated the syringe pump's dispense rates for three different disposable syringe volumes (5 mL, 10 mL, and 12 mL) at dispense speed setting 1 (Fig 1D), speed setting 5 (Fig 1E), and speed setting 9 (Fig 1F) by dispensing water gravimetrically onto a calibrated weighing balance. The plots of volume dispensed vs. pumping time elapsed show highly linear profiles for all three syringes, allowing us to calculate the volumetric dispense rates from the slope of each graph. Dispense rates ranged from 21.42 $\mu$L/min–1.82 mL/min for the 5 mL syringe, 32.86 $\mu$L/min–2.42 mL/min for the 10 mL syringe, and 32.93 $\mu$L/min–2.67 mL/min for the 12 mL syringe. Each dispense rate measurement was repeated three times. The small standard deviations and goodness of the linear fits ($R^2$ > 0.9936) suggest that this syringe pump provides a constant liquid dispensing rate at each speed setting. Slower or faster flow rates can be achieved by changing the gear ratio of the syringe pump or by modifying the syringe holder to fit smaller or larger volume syringes.

## A LEGO® orbital shaker/reaction agitator

Orbital shakers are commonly used to agitate reaction mixtures, ranging in size from small benchtop devices to table-sized enclosures with heating or refrigeration. Prices for commercial benchtop orbital shakers range from ∼80 USD (used) to ∼5,000 USD (new), depending on manufacturer, size, and features, based on prices observed in various online marketplace listings in November 2024. In general, orbital shakers constitute a platform to which vessels are affixed, which is rotated in a circular orbit. While nutating (3D) rotation modes are available in some models for certain applications, rotation is parallel to the benchtop surface in most orbital shakers. The shaking mechanism typically involves a stationary base with four points of attachment to the corners of the platform, each with a short offset (the length of which defines the rotary diameter, which is often near ∼2 cm in benchtop models) from four vertically-attached axles mounted to bearings. One or more of these axles is driven by a motor, which controls the speed (ranging from 0-300 rpm in most benchtop models) of rotation. Orbital shakers are often used in cell culture [55,56] and protein expression [57], but can also be used to agitate reactions in high-throughput chemical synthesis [58].

We designed an all-LEGO® orbital shaker (Fig 2) that is less costly and far more customizable than most dedicated laboratory orbital shakers. It can be assembled (instructions are provided in the Supplemental Information) from 458 LEGO® Technic™ parts at a cost of ∼55 USD (based on lowest prices on bricklink.com, June 2024). The drive system is powered by a

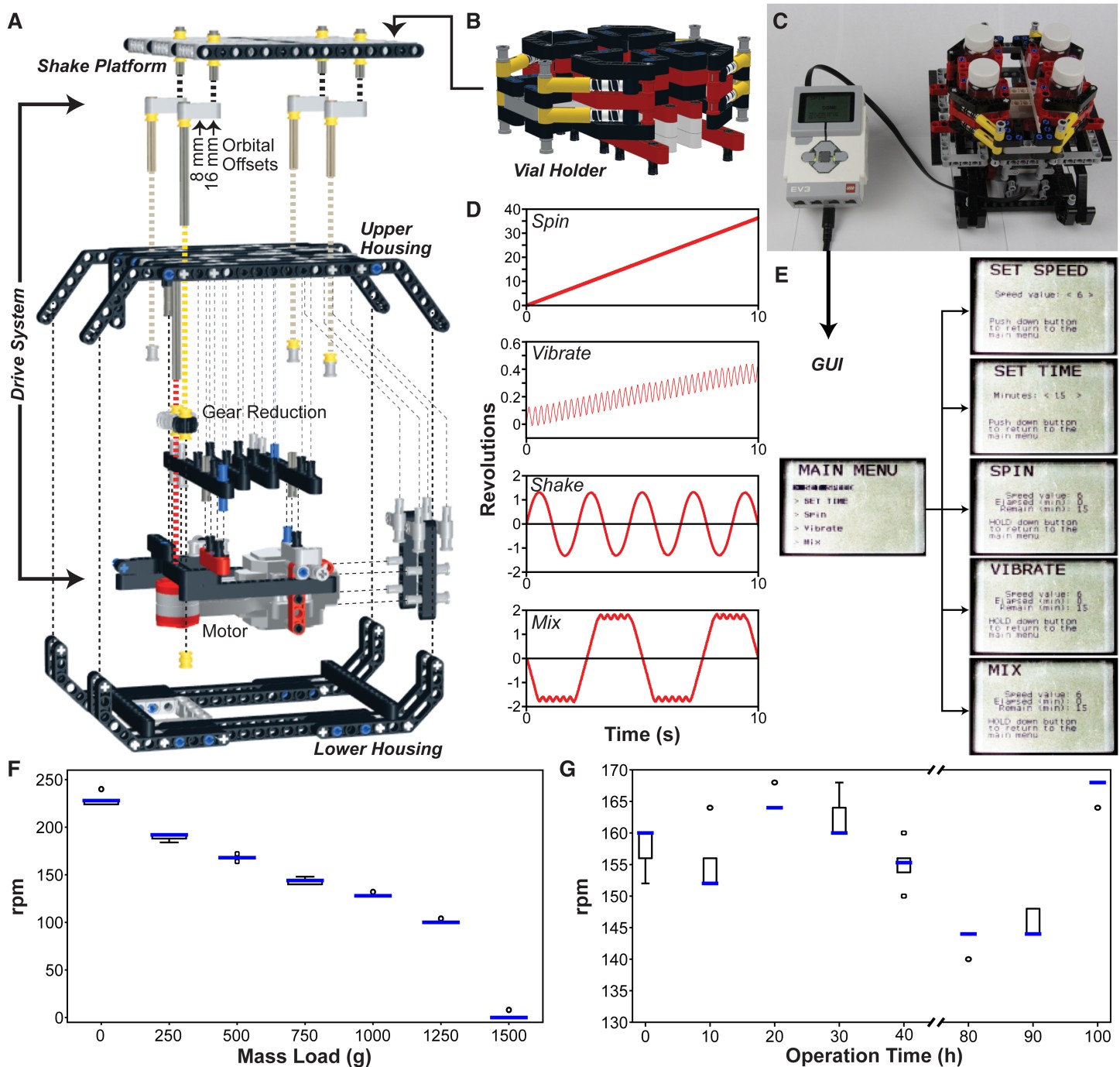

**Fig 2. A LEGO® orbital shaker/reaction agitator.** (**A**) Exploded oblique view of the LEGO orbital shaker and key subcomponents. (**B**) A vial holder designed to mount on the shake platform. (**C**) Shake profiles (revolutions vs. time) of a standard unidirectional, constant-speed orbital shaking pattern ("Spin"), as well as custom-programmed bidirectional shake profiles with low-amplitude vibrations ("Vibrate"), high-amplitude shaking ("Shake"), and a hybrid mixing sequence that combines low- and high-amplitude shaking ("Mix") modes. Positional data corresponding to clockwise (positive) and counterclockwise (negative) revolutions was captured directly from the wire used for communication between the EV3 brick and the orbital shaker's motor. (**D**) Photograph of the fully assembled LEGO orbital shaker. (**E**) Flowchart of the GUI menu. (**F**) Box plots of revolutions per minute (rpm) observed with increasing calibration weights loaded in 250 g intervals. The rpm observations at each mass load were recorded five times. (**G**) Box plots of revolutions per minute (rpm) observed in 10h intervals operating at maximum motor speed for 100 h when loaded with 4 scintillation vials, each filled with 20 mL of water. The rpm observations at each time point were recorded five times.

LEGO® EV3 Large Servo Motor, which is specified to reach speeds of up to 170 rpm, a running torque of 20 N/cm and a stall torque of 40 N/cm. The design (Fig 2A) can be subdivided into five sub-assemblies: lower housing, motor and gear reduction, upper housing, shake platform, and vial holder (Fig 2B). As-built (Fig 2C), the 15 × 14 cm shake platform agitates the vial holder, which can hold up to four vials ranging in size from 20-mL scintillation vials to 50-mL centrifuge tubes. With an empty vial holder, the shake platform can rotate at observed frequencies of 60–220 rpm at an orbital diameter of 32 mm.

To showcase the customizability of the orbital shaker, we demonstrated (Fig 2D) not only a standard fixed-rate unidirectional rotation, but also a low-amplitude vibrating motion, large-amplitude bidirectional shaking, and a hybrid mix pattern combining these bidirectional rotating and vibrating motions. The rotation vs. time profiles (Fig 2D) for each shake mode were captured electronically, directly from the data being exchanged by the EV3 and the motor. Different drive patterns can be selected through the GUI menu, which enables the user to set the shaking mode (spin, vibrate, or mix), rate, and time (Fig 2E).

The LEGO®-based orbital shaker is able to agitate up to 1.25 kg of mass (Fig 2F) and operate continuously for at least 100 h (Fig 2G). It is noteworthy that the modular nature of the LEGO® Technic™ platform also allows many parameters to be adjusted. For example, the LEGO® Technic™ liftarm beams that define the rotational offset of the drive system can be varied at intervals of ~8 mm (defined by the hole-to-hole distance of the Technic™ beams), enabling orbital diameters intervals of 2x this offset (16 mm, 32 mm, 48 mm, etc.). Similarly, the gear reduction between the motor and the platform can be varied with off-the-shelf LEGO® gears to customize its maximum frequency. The shake platform is based on a simple grid of 5 cm × 7 cm rectangular studless Chassis Frame Liftarm Beams, and can therefore be expanded or contracted in dimensional intervals proportional to these platform subunits (e.g., 10 cm × 7 cm, 5 cm × 14 cm, 15 cm × 14 cm, 20 cm × 21 cm, etc.). Our vial holder (Fig 2B) represents a proof-of-concept demonstration that LEGO® parts can be employed to custom-build vessel holders as needed; for example, a flat LEGO®-based platform could be introduced to accommodate a petri dish or beaker. Alternatively, vessels could be affixed to the platform by a variety of simple methods such as adhesive tape, hook-and-loop fasteners, or rubber bands. By giving users full control over the drive system's mechanics and controls, the LEGO® orbital shaker is more customizable than most orbital shakers on the market.

## A LEGO® microcentrifuge

A centrifuge is a tool used in research laboratories to separate materials of different density through centrifugal force [59]. A centrifuge is powered by a strong motor, which drives a rotor that holds centrifuge tubes to spin at high rotational speeds. For liquid suspensions of particles, centrifugation can accelerate the sedimentation of solid pellets at the bottom of the centrifuge tube, facilitating the isolation of either the pellet or the supernatant. The rotor typically holds an even number of centrifuge tubes, which must be filled with equal masses of material to maintain rotor balance and avoid damage to the instrument. In a fixed-angle centrifuge, the centrifuge tubes can be held at an angle that is perpendicular or parallel to the axis of rotation, or any angle in between [60]. A microcentrifuge functions by the same mechanisms, but is used for smaller sample volumes (≤ 1 mL). Microcentrifuges are predominantly used in biomedical research, such as DNA extraction [61], isolation of small samples of cultured cells and tissue [62], and separation of plasma from blood [63], so some microcentrifuges incorporate cooling functions to increase cell viability. The rotational speed of a commercial microcentrifuge ranges from 300 to 10000 rpm with an estimated cost that ranges

between ~100 USD (used) to ~5,000 USD (new), based on prices observed in various online marketplace listings in November 2024.

We designed an all-LEGO® low-speed microcentrifuge (Fig 3), which can be assembled (instructions are provided in Supplemental Information) with 252 pieces at a cost of ~52

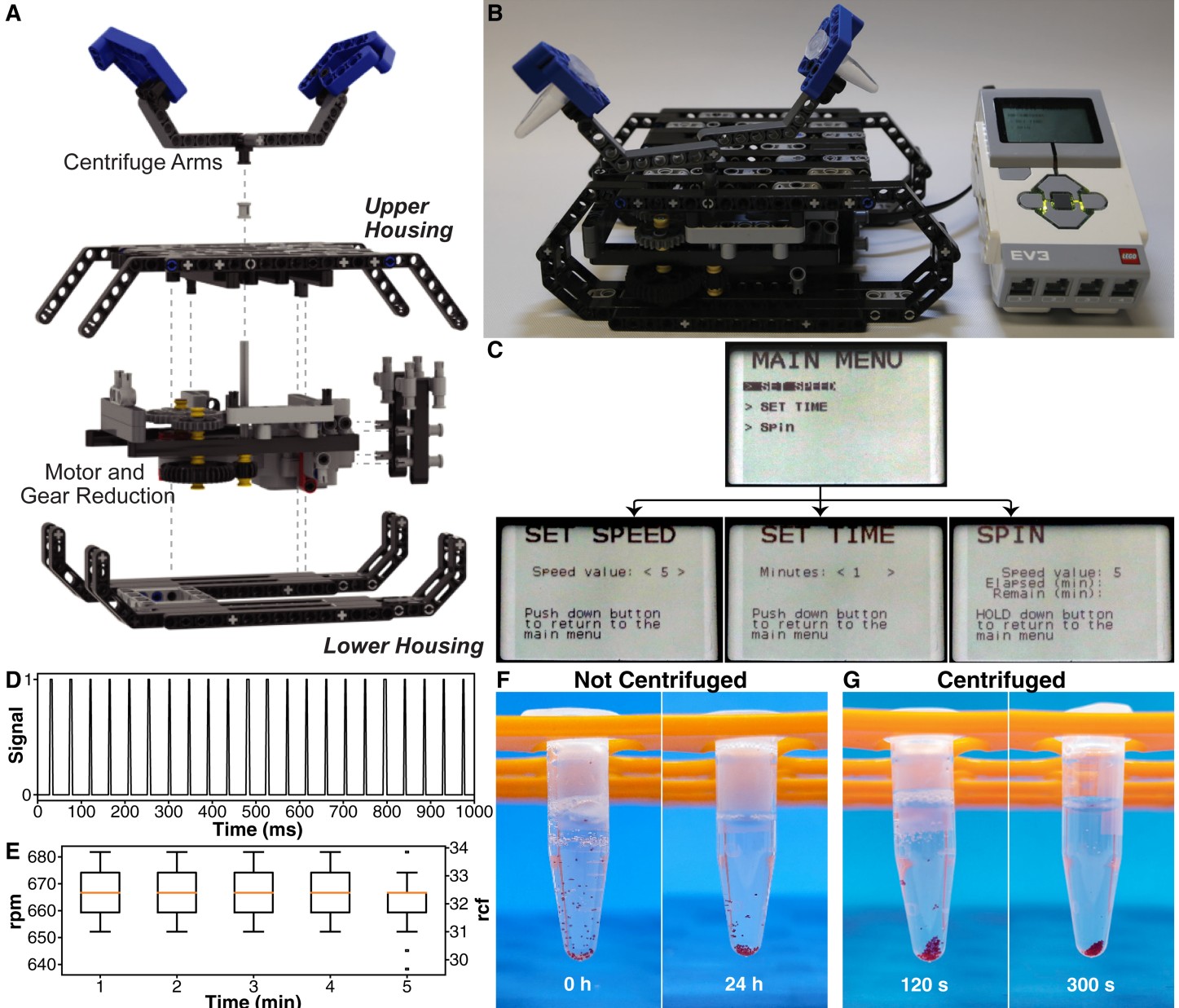

**Fig 3. A LEGO® microcentrifuge.** (**A**) Exploded oblique view of the four main subassemblies of the LEGO® microcentrifuge and key subcomponents. (**B**) Photograph of the fully assembled LEGO® microcentrifuge. (**C**) Graphical User Interface (GUI) menu flowchart. (**D**) Example of one second of data output by a photogate used to detect revolutions of the rotor at maximum centrifugation speed. (**E**) Box plot of revolutions per minute (rpm) and relative centrifugal force (rcf) measured at one-minute intervals during 5-min continuous runs. (**F**) Photographs of a microcentrifuge tube filled with polyacrylamide microgels (200 × 200 × 100 μm) after 0 h (left) and 24 h (right) of sedimentation under gravitational force. (**G**) Photographs of a microcentrifuge tube filled with polyacrylamide-based microgels (200 × 200 × 100 μm) after 120 s (left) and 300 s (right) of centrifugation at ~630 rpm / 30 rcf.

USD (based on lowest prices on bricklink.com, June 2024). The microcentrifuge has four sub-assemblies (Fig 3A): a lower housing, motor and gear reduction, upper housing, and a pair of 6.5-cm centrifuge arms, which can each hold one 2 mL microcentrifuge tube. The base design is similar to that of the LEGO® orbital shaker, making it easily reconfigurable, with a drive system powered by the same LEGO® EV3 Large Servo Motor. As-built (Fig 3B), the dimensions of the microcentrifuge are 20 × 14 × 8 cm. The GUI (Fig 3C) on board the EV3 brick allows the user to set the speed on a scale of 1–9, centrifugation time, and to start and monitor the status of centrifugation.

We measured the rotational speed of the microcentrifuge using an electronic photogate system. At the highest motor speed setting of 9, the photogate system detected 22 centrifuge arms per second (Fig 3D), corresponding to approximately 11 rotations per second or 660 revolutions per minute (rpm). To demonstrate that the rotational rates remained constant, we calculated the instantaneous rpm at 1-min intervals for 5 min (Fig 3E) by recording the temporal distance between every two photogate readings (to account for two centrifuge arms) for 10 s each. The average spin rate over this 5-min period was approximately 667 rpm, corresponding to approximately 32 units of relative centrifugal force (rcf).

To demonstrate an application of the all-LEGO® low-speed microcentrifuge, we fabricated polyacrylamide-based microgels of $200 \times 200 \times 100$ $\mu$m dimensions. Photographs in Fig 3F show the microgel suspension without centrifugation after 0 s and 24 h of gravitational settling time, compared with the same sample after 120 s and 300 s of centrifugation (Fig 3G) at the highest speed setting of the device used (approximately 630 rpm / 30 rcf). The results show that these micron-scale gel particles, which could not be separated in 24 h under gravitational force, were successfully separated within 5 minutes of continuous centrifugation with the LEGO® microcentrifuge. Similar to the orbital shaker / reaction agitator, the microcentrifuge can be further adjusted to customize its mechanics and controls if needed. For example, the microcentrifuge tube holder can be adjusted to accommodate smaller or larger volumes, while the gear reduction and/or centrifuge arm length can be adjusted to customize speed and rcf.

## Device validation in CaCO₃ microparticle synthesis

To further demonstrate the feasibility of using LEGO®-based laboratory equipment for scientific research experiments, we employed all three machines reported here (syringe pump, orbital shaker/reaction agitator, microcentrifuge) to synthesize and isolate $CaCO_3$ microparticles, comparing the results with an analogous procedure performed with commercial lab equipment. We modified a literature protocol[51] by using a syringe pump to dispense 2 mL of aqueous 0.33 M $Na_2CO_3$ solution into a 20 mL scintillation vial containing 2 mL of aqueous 0.33 M $CaCl_2$ solution, which was agitated at ~60 rpm on a conventional (22 mm orbital diameter, Fig 4A) or LEGO®-based (32 mm orbital diameter, Fig 4B) orbital shaker during liquid transfer, then agitated at ~210 rpm for 10 min afterwards. The resulting $CaCO_3$ microparticles could be separated successfully after 5 min of centrifugation at maximum speed in both a conventional microcentrifuge (reported 6000 rpm / 2000 rcf, Fig 4C) and the LEGO®-based microcentrifuge (measured 637 rpm / 30 rcf, Fig 4D) with similar yields in each case. The isolated yields of $CaCO_3$ from 1 mL of crude reaction mixture were 16.60 ± 0.22 mg and 16.41 ± 0.31 mg for samples processed with the conventional and LEGO®-based microcentrifuges, respectively, averaged over 6 trials.

The $CaCO_3$ microparticles were further characterized by dynamic light scattering (DLS, Fig 4E) and scanning electron microscopy (SEM, Fig 4F–H). With unidirectional orbital

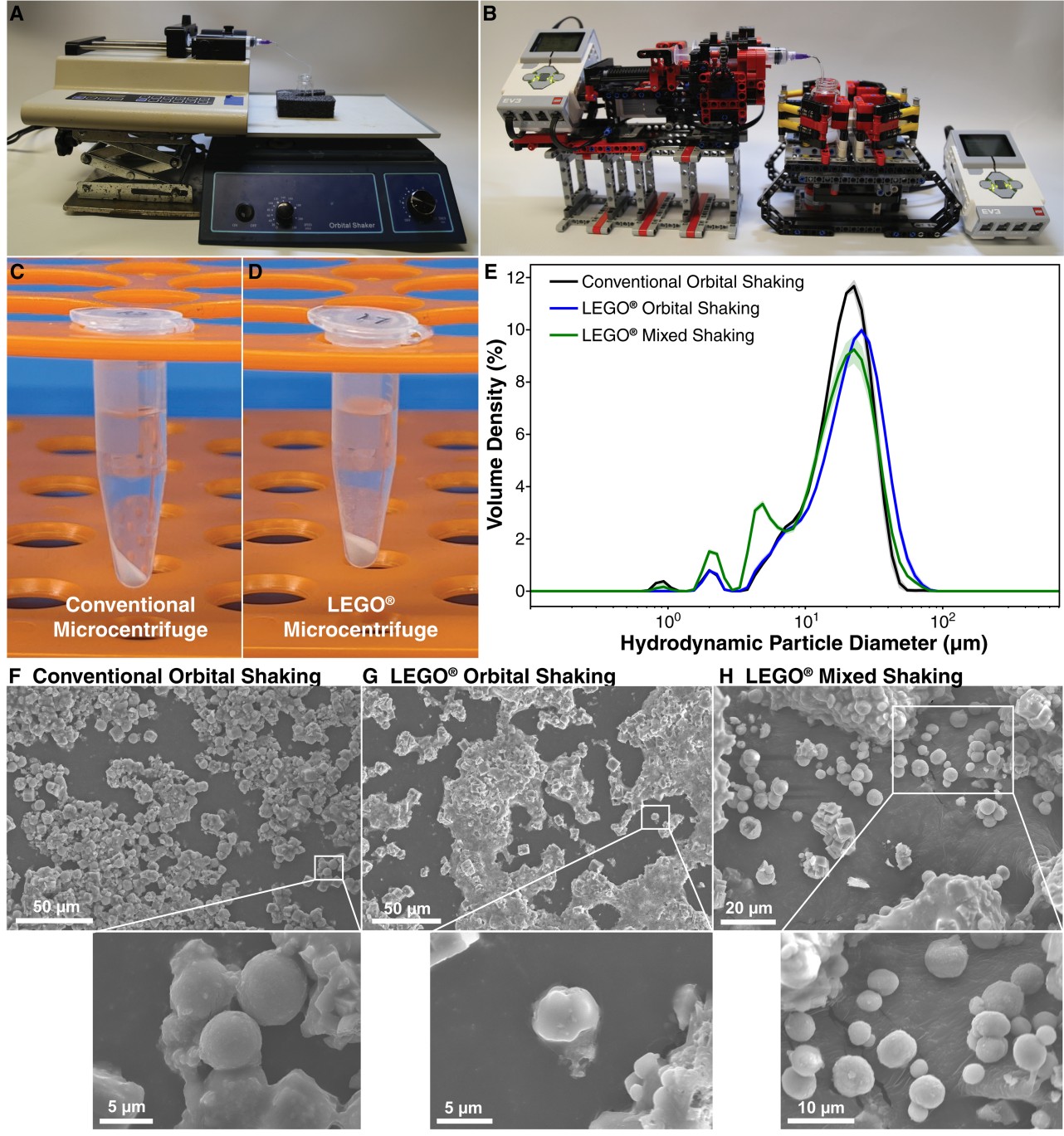

**Fig 4. Comparison of CaCO₃ microparticle synthesis with conventional and LEGO®-based lab equipment.** (**A**) Photograph of the reaction setup with a conventional syringe pump and orbital shaker. (**B**) Photograph of the CaCO₃ reaction setup with LEGO®-based syringe pump and orbital shaker. (**C**) Photograph of CaCO₃ microparticles separated in a conventional microcentrifuge (6000 rpm/2000 rcf, 16.60 ± 0.22 mg yield). (**D**) Photograph of CaCO₃ microparticles separated in a LEGO®-based microcentrifuge (~630 rpm/30 rcf, 16.41 ± 0.31 mg yield). (**E**) The mean of volume-weighted distributions of hydrodynamic particle diameter for suspensions of CaCO₃ microparticles prepared with conventional lab equipment (black), as well as LEGO®-based lab equipment with standard orbital shaking (blue) and a custom-programmed mixed shaking mode (green). The shaded fill regions indicate the standard deviation of the mean over five trials. (**F**) SEM images of CaCO₃ microparticles synthesized with conventional lab equipment. (**G**) SEM images of CaCO₃ microparticles synthesized with LEGO®-based lab equipment in orbital shake mode. (**H**) SEM images of CaCO₃ microparticles synthesized with LEGO®-based lab equipment in the custom-programmed mixed shake mode.

shaking at a constant rate of 210 rpm, the size distribution of particles obtained from the conventional and LEGO®-based equipment are remarkably similar. The full width at half maximum (FWHM) of volume-weighted hydrodynamic diameter range in microparticles synthesized using conventional lab equipment was 19.59 $\mu$m with a modal diameter of 22.60 $\mu$m, which compared similarly to the 22.25 $\mu$m FWHM and 25.68 $\mu$m modal diameter of particles obtained with LEGO®-based equipment. The particles were morphologically similar in both cases (Fig 4F-G), comprising mostly faceted microcrystals with a minority population of spherical microparticles.

To demonstrate that customization of the shake profile can modulate the experimental outcome, we synthesized a third batch of $CaCO_3$ microparticles with 10 min of bidirectional shaking and vibrating motions in the "Mix" shaking mode, accessible only in the LEGO®-based orbital shaker. The custom shaking mode yields more polydisperse $CaCO_3$ microparticles, giving rise to new enhanced peaks in the DLS data (Fig 4E) corresponding to smaller-than-average particle populations of ~2 $\mu$m and ~5 $\mu$m. SEM images of these $CaCO_3$ microparticles (Fig 4H), inaccessible by conventional orbital shaking, revealed that a greater proportion of spherical morphologies were obtained relative to the syntheses agitated by standard unidirectional rotation. These results demonstrate that the customization afforded by the LEGO® platform may provide access to a wider diversity of experimental outcomes in some cases.

## Stress testing of LEGO®-based Lab equipment

We performed some some stress-testing experiments to evaluate how LEGO®-based lab equipment might withstand conditions such as elevated temperature or repeated and prolonged real-world use. These stress tests may help potential users evaluate the suitability of the equipment for their application by clarifying the limits of its capabilities.

**Effect of elevated temperature.** The syringe pump, orbital shaker, and microcentrifuge were all operated at ~ 45 ± 5 ° C in an oven to evaluate how high laboratory temperatures might impact performance. The syringe pump operated successfully under these conditions, with linear dispense profiles and rates of 22.13 $\mu$L/min–1.92 mL/min (5 mL), 29.27 $\mu$L/min–2.44 mL/min (10 mL), and 29.08 $\mu$L/min–2.69 mL/min (12 mL), which were reasonably consistent with those observed at ~ 21 ° C (Fig S1, Supplemental Information). The orbital shaker was operated at maximum speed for 10 h inside the oven, where it maintained a consistent average of 220 ± 0.94 rpm over 3 runs with an empty vial holder. Only the LEGO® microcentrifuge suffered performance loss in the oven; a Technic™ bush dislodged from the axle after 2 h of continuous maximum-speed operation, causing the centrifuge arms to stop rotating due to gear misalignment, and continued to exhibit erratic rotation (Fig S2, Supplemental Information) even after re-aligning the gears.

**Effect of prolonged operation.** The long-term repeated cycling of the orbital shaker and microcentrifuge may make them prone to wear, so we performed some wear tests on these devices during periods of prolonged operation. We operated the orbital shaker at maximum speed for 100 h when loaded with four scintillation vials, each filled with 20 mL of water. Measuring rotational speed in 10 h intervals, we found that the speed remained reasonably consistent at an average of 155 ± 8 rpm for 100 h (Fig 2G). However, we noticed some deviations in frequency and occasional squeaking sounds after 50 h of continuous operation due to an overworked motor; these issues resolved after allowing the motor to rest for >24 h, suggesting that continuous operation should be limited to ~48 h. The microcentrifuge was operated at maximum speed for 7 h while loaded with two 1.5 mL tubes, each filled with 1 mL of water. The rotational speed remained consistent at an average of 554 ± 34 rpm (22.31 ± 0.08 rcf) for

6 h, then began to exhibit more erratic rotation after 7 h. Although the centrifuge continues to operate after 7 h, the photogate begins to detect the arms at irregular time intervals, (Fig S3, Supplemental Information).

**Failure analysis and mitigation.** We analyzed the failure modes of the microcentrifuge, since it was most susceptible to performance loss in both temperature and prolonged operation stress tests. The high-temperature failure was attributed to misalignment of the axles, gears, and bushings in the drive system, most likely caused by a combination of thermal expansion and vibrations from the motor. The erratic rotations that emerged over prolonged operation were attributed to wear of the plastic parts. Inspection of the Technic™ axles, bushes, liftarms, and connectors of the drive system revealed a dust of plastic shavings generated by wear-induced damage, while the surface roughness of the rotating axle was apparent under magnification (Fig S4, Supplemental Information). Although the worn LEGO® parts can be replaced easily at low cost (∼ $0.06 USD) to restore the function of the microcentrifuge, we further reasoned that it may be possible to protect these wear points with a lubricant. After applying mineral oil to the gear system's coupling of the rotating axle and liftarm (identified as the most significant wear point), we operated the LEGO® microcentrifuge again for 7 h. The lubricant increased the initial average spin rate from $667 \pm 9$ rpm ($32.36 \pm 0.01$ rcf) to $693 \pm 9$ rpm ($34.98 \pm 0.01$ rcf) and reduced visible wear to the parts, but did not completely preserve the function of the pristine device, since rotation again became more erratic after 7 h (Fig S5, Supplemental Information).

**Rebuild consistency.** Upon disassembly, exchanging or replacing parts, and re-assembly, four different builds of the microcentrifuge were not perfectly consistent in performance; four different average maximum rpm rates ranging from 600–700 rpm (Fig S6, Supplemental Information). After swapping different motors and EV3 bricks among different builds, we were able to attribute these differences in maximum rotational rate to performance variation among different individual motors and EV3 controllers. These experiments revealed a limitation that LEGO®-based lab equipment would need to be re-calibrated for each individual build or parts replacement for users to maintain accurate operational speed settings.

We conclude that the service life of LEGO®-based lab equipment is likely to be significantly lower than dedicated lab equipment, but note that simple solutions such as low-cost part replacement or lubricant can help mitigate these issues.

## Cost and labor analysis of LEGO®-based lab equipment

In addition to customizability, the cost and reconfigurability of the LEGO® Technic™ system presents considerable economic advantages. The Venn diagram in Fig 5 visualizes how many pieces are shared between the LEGO®-based syringe pump, orbital shaker/reaction agitator, and microcentrifuge, as well as their costs based on lowest available pricing on bricklink.com in June of 2024. While the cost to assemble all three devices in parallel from 1215 pieces is already quite low (<174 USD) in comparison to most conventional equipment, this cost is further lowered to < 83 USD when the machines are constructed in series from 687 pieces, since all three machines share 143 pieces in common and an additional 241 pieces are shared by at least two machines. The full list of parts and cost is available in Table S1, Supplemental Information. This ability to re-purpose any part of an old machine for the construction of a new one is not possible with most conventional lab equipment, which is already more expensive – even on the lowest end of the price range – than the corresponding LEGO®-based counterparts.

Although the LEGO® equipment clearly offers cost savings compared to conventional laboratory equipment, there is a labor trade-off for design, procurement, and assembly. We

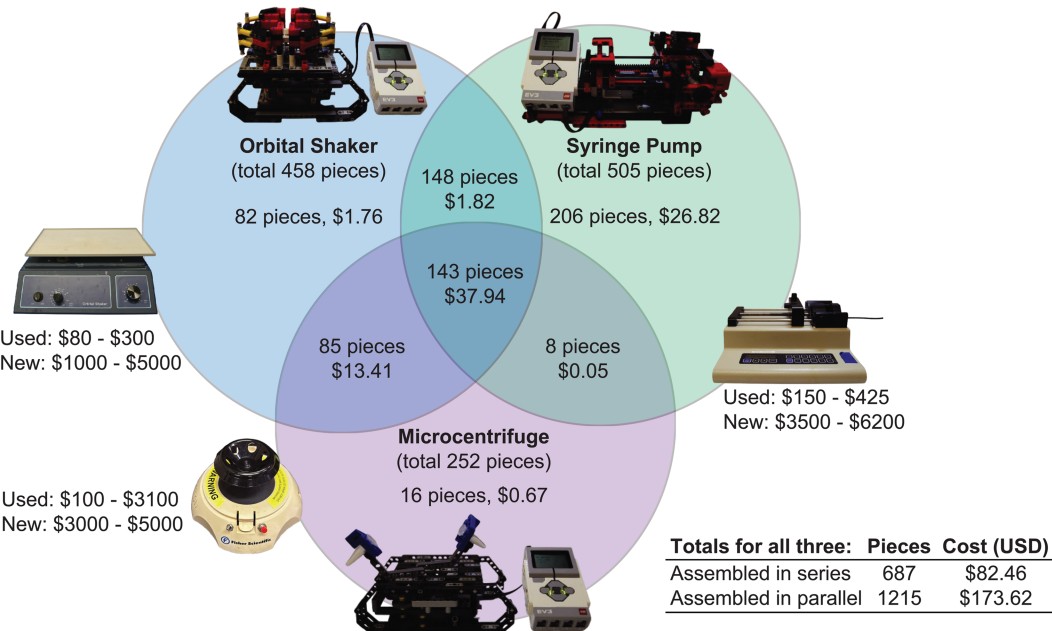

| | Pieces | Cost (USD) |
|---|---|---|
| **Totals for all three:** | | |
| Assembled in series | 687 | $82.46 |
| Assembled in parallel | 1215 | $173.62 |

**Fig 5. A visual comparison of the number and cost of LEGO® pieces employed to construct an orbital shaker, syringe pump, and microcentrifuge, in comparison with analogous conventional laboratory equipment.** A complete list of parts can be found in the Supplemental information.

designed and programmed each device in ~100 h of focused work, while ~40 min of procurement time was sufficient to order all parts required to assemble the three devices. Assembly times will depend on experience, but likely range from ~1–6 h in most cases. We found that it took an experienced builder ~1 h to assemble the device with a minimum number of steps (the microcentrifuge), while it took a novice builder ~5 h to complete the assembly of the device with the maximum number of steps (the syringe pump).

## Discussion

LEGO® products have been widely used in K-12 STEM education [64], where they have been shown to be favorable and effective in teaching science and engineering concepts to students [45,49,65,66], and increase student motivation by providing autonomy and opportunity to develop technological competence [67]. For example, LEGO®-based systems have been used to teach robotics [68,69], hardware and software skills [70], mechatronics and programming skills [49], nanoscale chemical structures and properties [71], gene sequencing [66], and even social-emotional learning [72] to elementary, middle, and high school students. Therefore, we envision the LEGO® laboratory equipment presented in this paper could be utilized for engaging hands-on lessons in K-12 STEM curricula to teach students hardware and software skills, as well as experimental biology and chemistry. However, in this report, our aim was to investigate the feasibility of using LEGO® laboratory equipment for real experimental research.

LEGO® systems provide a way to build custom lab equipment with a low up-front investment, and the parts can be assembled, disassembled, reconfigured and reused as needed, while all of the components and controls are integrated in a single unified and user-friendly

platform. Inspired by these advantages, we designed and constructed three all-LEGO® laboratory automation machines – a syringe pump for liquid dispensing, an orbital shaker for reaction agitation, and a microcentrifuge for separation – to evaluate the cost and feasibility of employing LEGO®-based equipment instead of conventional commercial laboratory equipment in experimental research. All three devices were validated to give similar results as commercial lab equipment in experiments to synthesize and isolate $CaCO_3$ microparticles. Furthermore, custom bidirectional shake profiles programmed in LEGO®*MINDSTORMS®* software yielded differences in microparticle size and morphology that were not accessible by conventional unidirectional shaking; thus the customizability of LEGO®-based equipment may offer unique advantages in some experiments. The three devices reported here are only representative examples; in principle, countless instruments can be constructed with LEGO® pieces to support scientific research.

Limitations to the LEGO®-based tools include higher labor demands and potentially shorter service lifetimes due to mechanical wear. We found that plastic parts adjacent to the gears (i.e., Technic™ liftarm, axle, and bush) were the most prone to wear, especially for the microcentrifuge where rapid rotations cause continuous scratching against the plastic surface. Although the optimum service life is only 6 h at maximum speed, the worn parts can be replaced at a cost of $0.06 USD, which restores the function of the microcentrifuge. Lubricant can also help prolong the service life of high-wear parts. We also observed that runtime limitations can arise from overworked motors over multiple days of stress testing, as well as the LEGO® MINDSTORMS® software which limits continuous operation to a maximum of 999 min.

As researchers have become more environmentally conscious, single-use plastics and energy-demanding instruments have been a rising issue in scientific research [73]. Re-using materials, resources, and instruments improves the sustainability, cost efficiency, and environmental impacts of scientific research [74]. For example, green-lab initiatives in institutions across the world have introduced resource-sharing platforms to help mitigate research waste and maximize equipment service life [12]. Some have found innovative ways of keeping aged equipment from landfills, such as Stanford University's annual lab swap event [75] and Los Alamos National Laboratory's recycling project [76]. In this report, we demonstrated that almost half as many pieces are needed to build our three LEGO® instruments in series compared to building them in parallel, indicating a high degree of reconfigurability and reusability, which can further help reduce plastic waste. The disassemble-and-reuse approach enabled by the LEGO® platform may provide a complementary strategy for reducing the waste, cost, and resource demands in scientific research. Overall, our results support the feasibility of using LEGO®-based tools for scientific research that conserves on budgetary and material resources.

## Acknowledgments

We acknowledge the Colorado Shared Instrumentation in Nanofabrication and Characterization (COSINC) for the use of the Hitachi SU3500 VP Scanning Electron Microscope and the Cressington 108 Auto/SE Sputter Coater. We acknowledge Professor Wil V. Srubar (University of Colorado Boulder) for access to the Malvern Panalytical Mastersizer 3000 and Matthew Fyfe and Aseem Visal for assistance with particle size analysis. We acknowledge Dr. Subhankar Mandal (ATLAS Institute, University of Colorado Boulder) for helpful technical discussions and assistance with microgel fabrication. We acknowledge Ariana Morales Garcia and Zoey Coffie for assistance in build time assessment.

## Supporting information

**S1 File. Supplementary Information. Figures S1–S6 and Table S1.**
(PDF)

**S2 File. Assembly instructions for LEGO®-based lab equipment.**
(PDF)

**S3 File. Time-lapse assembly video of the LEGO®-based syringe pump.**
(MP4)

**S4 File. Time-lapse assembly video of the LEGO®-based orbital shaker.**
(MP4)

**S5 File. Time-lapse assembly video of the LEGO®-based microcentrifuge.**
(MP4)

**Accession codes** The GUI codes for the LEGO® MINDSTORMS® EV3 can be found at
https://github.com/diane-jung/LEGO_lab-equipment.git.

## Author contributions

**Conceptualization:** Diane N. Jung, Kailey E. Shara, Carson J Bruns.

**Data curation:** Diane N. Jung.

**Formal analysis:** Diane N. Jung, Carson J Bruns.

**Funding acquisition:** Carson J Bruns.

**Investigation:** Diane N. Jung, Kailey E. Shara.

**Methodology:** Diane N. Jung, Kailey E. Shara.

**Project administration:** Carson J Bruns.

**Resources:** Carson J Bruns.

**Software:** Diane N. Jung, Kailey E. Shara.

**Supervision:** Carson J Bruns.

**Validation:** Diane N. Jung, Kailey E. Shara.

**Visualization:** Diane N. Jung.

**Writing – original draft:** Diane N. Jung, Kailey E. Shara, Carson J Bruns.

**Writing – review & editing:** Diane N. Jung, Carson J Bruns.

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
