## [Decision Letter · Decision Letter 0]

21 Mar 2025

PONE-D-25-08317LEGO® as a versatile platform for building reconfigurable low-cost lab equipmentPLOS ONE

Dear Dr. Bruns,

Thank you for submitting your manuscript to PLOS ONE. After careful consideration, we feel that it has merit but does not fully meet PLOS ONE’s publication criteria as it currently stands. Therefore, we invite you to submit a revised version of the manuscript that addresses the points raised during the review process.

We look forward to receiving your revised manuscript.

Kind regards,

Yuliang Zhang, Ph.D.

Academic Editor

PLOS ONE

“U.S. National Science Foundation (NSF) Award No. 2222952. Collaborative Research: FW-HTF-R: RoboChemistry: Human-Robot Collaboration for the Future of Organic Synthesis.”

Reviewers' comments:

Reviewer's Responses to Questions

**Comments to the Author**

1. Is the manuscript technically sound, and do the data support the conclusions?

Reviewer #1: Yes

Reviewer #2: Yes

2. Has the statistical analysis been performed appropriately and rigorously? 

Reviewer #1: N/A

Reviewer #2: Yes

3. Have the authors made all data underlying the findings in their manuscript fully available?

Reviewer #1: Yes

Reviewer #2: No

4. Is the manuscript presented in an intelligible fashion and written in standard English?

Reviewer #1: Yes

Reviewer #2: Yes

5. Review Comments to the Author

Reviewer #1: Jung et al. present the LEGO version of three corner-stone wet-lab devices, and explore their limitations and capabilities. Overall, a well-thought out and written account, and applicable and engaging to the general audience. The manuscript can be published as-is, however making the following changes would enhance the project:

- Make a time-lapse assembly video of each of the devices.

- Investigate environmental factors on the performance and stability of the devices, e.g. humidity, temperature, dust, etc.

- Describe/discuss in greater detail how the different devices could be improved/stabilized. E.g. where/on which part would the use of adhesives improve the device stability, albeit at the cost of reconfigurability; which part could be replaced with metallic or metalized 3D printed equivalent and lead to improve device, etc.

Reviewer #2: In the article “LEGO as a versatile platform for building reconfigurable low-cost lab equipment”, the authors have described how compact, multiunit, modular configuration of lab equipment can reduce the cost effectiveness, increase efficiency. They have demonstrated the feasibility of this concept for a syringe pump, orbital shaker, and a microcentrifuge with their corresponding graphical user interface for calcium carbonate microparticle synthesis and isolation.

The authors have demonstrated fluid injection using their modular syringe pump, but it would be nice if the authors can provide a rational on why syringe pumps are utilized instead of compact micropressure pumps (like Bartels micropump, etc) which are inexpensive as well and that are smaller, pulsation free and have the potential to have multiple of in a limited space.

The authors have shown that they can operate at 21 ul/min, are there any options in the syringe pump that would allow for lower flow rates?

Further explanation on the size limits of containers/tubes/vessels that can be utilized in the orbital shaker would be appreciated.

The mix shaking mode demonstrated by the authors is impressive.

Further explanation on how these instruments can be extended to other applications, and wear and tear assessment of these plastic components is strongly recommended, as they help other groups determine if its worthy to invest in such resources or if they can procure commercially available alternatives.

6. PLOS authors have the option to publish the peer review history of their article (what does this mean?). If published, this will include your full peer review and any attached files.

Reviewer #1: No

Reviewer #2: **Yes: **Sankar Raju Narayanasamy

---

## [Author Response · Author response to Decision Letter 1]

5 Jun 2025

Responses to reviewer comments are attached as a PDF, but copied below as plain text.

5 June 2025

Dear Dr. Zhang,

We thank you for inviting us to submit a revised version of our manuscript entitled “LEGO® as a versatile platform for building reconfigurable low-cost lab equipment” to PLOS ONE after receiving feedback from two peer reviewers. We have responded to editorial and reviewer comments in bold text below and revised the manuscript accordingly where applicable. We look forward to hearing back from you.

Sincerely,

Carson Bruns

Assistant Professor

ATLAS Institute / Rady Department of Mechanical Engineering

University of Colorado Boulder

RESPONSES TO EDITORIAL COMMENTS

PONE-D-25-08317

LEGO® as a versatile platform for building reconfigurable low-cost lab equipment

PLOS ONE

Dear Dr. Bruns,

Thank you for submitting your manuscript to PLOS ONE. After careful consideration, we feel that it has merit but does not fully meet PLOS ONE’s publication criteria as it currently stands. Therefore, we invite you to submit a revised version of the manuscript that addresses the points raised during the review process.

We look forward to receiving your revised manuscript.

Kind regards,

Yuliang Zhang, Ph.D.

Academic Editor

PLOS ONE

We have tried to align with PLOS ONE style guidelines in our manuscript.

“U.S. National Science Foundation (NSF) Award No. 2222952. Collaborative Research: FW-HTF-R: RoboChemistry: Human-Robot Collaboration for the Future of Organic Synthesis.”

The statement, “The funders had no role in study design, data collection and analysis, decision to publish, or preparation of the manuscript” is true and correct. This amended Role of Funder statement was added to the financial disclosure.

1. Is the manuscript technically sound, and do the data support the conclusions?

Reviewer #1: Yes

Reviewer #2: Yes

2. Has the statistical analysis been performed appropriately and rigorously?

Reviewer #1: N/A

Reviewer #2: Yes

3. Have the authors made all data underlying the findings in their manuscript fully available?

Reviewer #1: Yes

Reviewer #2: No

4. Is the manuscript presented in an intelligible fashion and written in standard English?

Reviewer #1: Yes

Reviewer #2: Yes

We thank the reviewers for validating the appropriateness of the manuscript in all cases except data availability in the case of Reviewer 2. However, Reviewer 2 did not specify what data is missing or unavailable in their comments. We humbly request further guidance on this issue and will be happy to provide our data in a format that the journal recommends.

5. Review Comments to the Author

Reviewer #1: Jung et al. present the LEGO version of three corner-stone wet-lab devices, and explore their limitations and capabilities. Overall, a well-thought out and written account, and applicable and engaging to the general audience.

We thank the reviewer for the positive comments on our paper, and for the suggestions below which we believe helped to further improve the manuscript.

The manuscript can be published as-is, however making the following changes would enhance the project:

- Make a time-lapse assembly video of each of the devices.

At the reviewer’s suggestion, we prepared time-lapse videos for each device assembly and included them as supporting information files.

- Investigate environmental factors on the performance and stability of the devices, e.g. humidity, temperature, dust, etc.

Inspired in part by this comment, we have added a new “Stress Testing” section to the revised manuscript that includes new experimental results on the effect of temperature on the equipment, though we do not have the equipment available to perform stress tests under controlled humidity or dust conditions.

- Describe/discuss in greater detail how the different devices could be improved/stabilized. E.g. where/on which part would the use of adhesives improve the device stability, albeit at the cost of reconfigurability; which part could be replaced with metallic or metalized 3D printed equivalent and lead to improve device, etc.

We thank the reviewer for the invitation to perform a more in-depth failure analysis. This new failure analysis has also been included in the new “Stress Testing” section of the revised manuscript. As described in this new section, we identified the microcentrifuge as the most susceptible equipment to wear and tear, so we focused on it for a case study in failure analysis. New photographs and micrographs (Fig. S3) reveal that wear-induced damage leads to erratic behavior after prolonged use. We also reported that this damage can be mitigated, but not avoided entirely, with the introduction of a lubricant, which also happens to increase the centrifugation speed. A comment in the manuscript also highlights that damaged parts can be replaced relatively quickly and easily at low cost.

Reviewer #2: In the article “LEGO as a versatile platform for building reconfigurable low-cost lab equipment”, the authors have described how compact, multiunit, modular configuration of lab equipment can reduce the cost effectiveness, increase efficiency. They have demonstrated the feasibility of this concept for a syringe pump, orbital shaker, and a microcentrifuge with their corresponding graphical user interface for calcium carbonate microparticle synthesis and isolation.

We thank the reviewer for taking the time to read and provide actionable critical feedback on our paper.

The authors have demonstrated fluid injection using their modular syringe pump, but it would be nice if the authors can provide a rational on why syringe pumps are utilized instead of compact micropressure pumps (like Bartels micropump, etc) which are inexpensive as well and that are smaller, pulsation free and have the potential to have multiple of in a limited space.

Bartels micropump and other micropumps may be viable tools for many laboratory settings where high volume throughput is not required; while these pumps can even be low in cost, they are not conducive to LEGO-based construction because they require special materials such as elastomeric membranes that are not available in the LEGO catalogue. We have re-written the introductory paragraph of the syringe pump section to include more discussion and a literature citation about different types of pumps, and acknowledged this limitation to the LEGO platform’s versatility in lines 161-167 of the revised manuscript.

The authors have shown that they can operate at 21 ul/min, are there any options in the syringe pump that would allow for lower flow rates?

We added a sentence to this effect in lines 202-204 of the manuscript. Lower flow rates can be achieved by changing the gear ratios and by reconfiguring the syringe holder to fit a smaller volume syringe.

Further explanation on the size limits of containers/tubes/vessels that can be utilized in the orbital shaker would be appreciated.

We clarified this flexibility in lines 241-256. The current orbital shaker vessel holder can hold as small as 20 mL scintillation vials to 50 mL centrifuge tubes. Due to the reconfigurability of LEGO bricks, the vessel holder can be redesigned to fit smaller or bigger beakers, or even a petri dish, for example.

The mix shaking mode demonstrated by the authors is impressive.

We thank the authors for the compliment about our innovative shake mode.

Further explanation on how these instruments can be extended to other applications, and wear and tear assessment of these plastic components is strongly recommended, as they help other groups determine if its worthy to invest in such resources or if they can procure commercially available alternatives.

This comment was echoed by Reviewer 1 so we have focused on a more detailed failure analysis in our revision. After continuous operation of the microcentrifuge for 7 h, we disassembled the device to assess wear and tear of the components. We designed the gear system such that the Technic axle passes through a hole in the Technic liftarm for stability. Friction from continuous rotation of the Technic axle causes wear and tear of the inner walls of the Technic liftarm holes and the outer part of the Technic axle, which has an X-shaped cross section. When the damaged axle was imaged under a microscope, we could visually distinguish the smooth surface of the undamaged part and roughness of the worn part. See the newly added Figure S4.

To assess the possibility of increasing service life by addressing these parts, we have applied mineral oil on the Technic axle/Technic liftarm to test if lubricant could address the issue of wear and tear. After 7 h of continuous operation at maximum speed, there was no visual damage to the device and the decline in device performance was less drastic compared to device functionality without lubricant. However, there was a decrease in average spin rate (~ 100 rpm) and an increase in spin rate standard deviation (~ 100 rpm), likely resulting from the photogate data showing erratic rotation. While there does appear to be damage even with the lubricant, it seems to increase the service life of the device. These new results are reported in the new “Stress Testing” section of the revised manuscript.

6. PLOS authors have the option to publish the peer review history of their article (what does this mean?). If published, this will include your full peer review and any attached files.

We elect to include peer review history published with our article.

---

## [Editor Report · Decision Letter 1]

9 Jun 2025

LEGO® as a versatile platform for building reconfigurable low-cost lab equipment

PONE-D-25-08317R1

Dear Dr. Bruns,

We’re pleased to inform you that your manuscript has been judged scientifically suitable for publication and will be formally accepted for publication once it meets all outstanding technical requirements.

Kind regards,

Yuliang Zhang, Ph.D.

Academic Editor

PLOS ONE